# Using referral rates for genetic testing to determine the incidence of a rare disease: The minimal incidence of congenital hyperinsulinism in the UK is 1 in 28,389

Daphne Yau[1]*, Thomas W. Laver[2], Antonia Dastamani[3], Senthil Senniappan[4], Jayne A. L. Houghton[5], Guftar Shaikh[6], Tim Cheetham[7], Talat Mushtaq[8], Ritika R. Kapoor[9], Tabitha Randell[10], Sian Ellard[2,5], Pratik Shah[3], Indraneel Banerjee[1], Sarah E. Flanagan[2]

1 Department of Paediatric Endocrinology, Royal Manchester Children's Hospital, Manchester, United Kingdom, 2 Institute of Biomedical and Clinical Science, University of Exeter Medical School, Exeter, United Kingdom, 3 Department of Paediatric Endocrinology, Great Ormond Street Hospital, London, United Kingdom, 4 Department of Paediatric Endocrinology, Alder Hey Children's Hospital, Liverpool, United Kingdom, 5 Genomics Laboratory, Royal Devon and Exeter NHS Foundation Trust, Exeter, United Kingdom, 6 Department of Paediatric Endocrinology, Royal Hospital for Children, Glasgow, United Kingdom, 7 Department of Paediatric Endocrinology, Royal Victoria Infirmary, Newcastle upon Tyne, United Kingdom, 8 Department of Paediatric Endocrinology, Leeds Children's Hospital, Leeds, United Kingdom, 9 Department of Paediatric Endocrinology, King's College London, London, United Kingdom, 10 Department of Paediatric Endocrinology, Nottingham Children's Hospital, Nottingham, United Kingdom

* dyau@qmed.ca

**Data Availability Statement:** The genotype data could be used to identify individuals and so cannot

## Abstract

Congenital hyperinsulinism (CHI) is a significant cause of hypoglycaemia in neonates and infants with the potential for permanent neurologic injury. Accurate calculations of the incidence of rare diseases such as CHI are important as they inform health care planning and can aid interpretation of genetic testing results when assessing the frequency of variants in large-scale, unselected sequencing databases. Whilst minimal incidence rates have been calculated for four European countries, the incidence of CHI in the UK is not known. In this study we have used referral rates to a central laboratory for genetic testing and annual birth rates from census data to calculate the minimal incidence of CHI within the UK from 2007 to 2016. CHI was diagnosed in 278 individuals based on inappropriately detectable insulin and/or C-peptide measurements at the time of hypoglycaemia which persisted beyond 6 months of age. From these data, we have calculated a minimum incidence of 1 in 28,389 live births for CHI in the UK. This is comparable to estimates from other outbred populations and provides an accurate estimate that will aid both health care provision and interpretation of genetic results, which will help advance our understanding of CHI.

## Introduction

Congenital hyperinsulinism (CHI), a disorder of heterogeneous aetiology characterised by dysregulated and inappropriate insulin secretion, is a significant cause of hypoglycaemia in

be made openly available. Access to data is open through collaboration. Requests for collaboration will be considered following an application to the Genetic Beta Cell Research Bank (https://www.diabetesgenes.org/current-research/genetic-beta-cell-research-bank/). Contact by email should be directed to the Lead Nurse, Dr Bridget Knight (b.a.knight@exeter.ac.uk).

**Funding:** SEF has a Sir Henry Dale Fellowship jointly funded by the Wellcome Trust (https://wellcome.ac.uk/funding) and the Royal Society (https://royalsociety.org/grants-schemes-awards/grants/) (Grant Number: 105636/Z/14/Z). The funders did not play any role in study design, data collection and analysis, decision to publish, or preparation of the manuscript.

**Competing interests:** The authors have declared that no competing interests exist.

neonates and infants with the potential for permanent neurologic injury. CHI can be transient or persistent. Transient CHI is usually associated with stress in the antenatal or perinatal period, such as intrauterine growth restriction or birth asphyxia, and typically resolves in the first few months of life, the majority by 6 months of age [1, 2, 3]. Individuals with persistent CHI continue to require medical therapy or surgery to stabilise blood glucose levels and up to 50–65% of patients in this group are without an identified mutation in any of the known CHI genes [4, 5]. Previous estimates of the incidence of CHI in outbred populations have ranged from 1 in 27,000 in Ireland to 1 in 50,000 in the Netherlands [6, 7]. The precise criteria for defining CHI however has varied between studies, at times have been unclear and has often not distinguished between transient and persistent forms [8, 9, 5]. Moreover, the incidence of CHI within the UK has yet to be determined. As accurate estimates are needed for both health care provision and research purposes, we have now calculated the minimal incidence of CHI within the UK based on referral rates to a central diagnostic genetic testing laboratory.

## Materials and methods

This study was approved by the North Wales Research Ethics Committee (517/WA/0327). Consent was not obtained as the data were analysed anonymously. The incidence of CHI for a region can be calculated from referral rates to a diagnostic laboratory and the annual birth rate. As not all cases will necessarily be referred for testing, this will reflect a minimum incidence for the region. The Exeter Molecular Genetics Laboratory, at the Royal Devon and Exeter Hospital (Exeter, UK) is the sole centre for clinical genetic testing for CHI in the UK. As the clinical care for patients with CHI in the UK is largely provided through two nationally commissioned, National Health Service-funded centres, the referral rate for genetic testing for this condition is high.

To calculate the minimal incidence of CHI over a 10-year period we identified all referrals for genetic testing for CHI from England, Wales, Scotland and Northern Ireland from 2007 to 2016. CHI was defined as having an inappropriate insulin and C-peptide measurement during hypoglycaemia in the clinical context of recurrent hypoglycaemia and high glucose infusion rate suggestive of hyperinsulinism [10, 11] diagnosed within the first 12 months of life. Although both insulin and C-peptide were measured, insulin was used as the primary biomarker with C-peptide as a supporting adjunct. All cases had laboratory glucose levels <3 mmol/L with a concurrent, unequivocally raised insulin level. Inclusion criteria were persistence of CHI beyond 6 months or requirement for pancreatectomy to manage hypoglycaemia. Persistence of CHI was defined as continuing need for medical therapy and/or dietary carbohydrate supplementation to prevent hypoglycaemia. The cut-off of 6 months was used to exclude cases of transient CHI, associated with environmental factors around the perinatal period, as most transient cases resolve within 6 months [1, 2, 3]. Data on the duration and management of CHI, including requirement for surgery, was obtained from the genetic testing request form. If this information was not provided at the time of testing, the referring physician was contacted to provide these details. Data on live births from 2007 to 2016 was obtained from the Office of National Statistics (https://www.ons.gov.uk/peoplepopulationandcommunity, accessed 27/07/19), National Records of Scotland (https://www.nrscotland.gov.uk/statistics-and-data/statistics, accessed 27/07/19) and the Northern Ireland Statistics and Research Agency (https://www.nisra.gov.uk/publications/birth-statistics, accessed 24/08/19). There were a total of 7,892,004 live births in the United Kingdom from 2007 to 2016 inclusive: 7,069,592 in England and Wales; 574,601 in Scotland and 247,811 in Northern Ireland. 95% Confidence intervals were calculated for the study population and the estimates from other countries where data were available for calculation (S1 Table).

## Results

Between 2007 to 2016, 278 UK patients (55% male) referred for genetic testing had biochemically-confirmed CHI persisting for more than 6 months or had undergone pancreatic surgery for CHI (74/278, 27%) (Fig 1). Median glucose was 1.8 mmol/L (interquartile range (IQR) 1.3, 2.3) with insulin 90 pmol/L (median; IQR 44,168) and C-peptide 700 pmol/L (median; IQR 374, 1224). CHI persisted beyond 12 months in 229 of these cases (82%). Consanguinity was reported in 24/278 patients (8.6%). Disease-causing variants in known CHI genes were identified in 154 (55%) individuals, of which mutations in the potassium ATP channel genes *ABCC8* and *KCNJ11* were the most common (115/154, 74%). The pick-up rate for disease-causing variants in patients with reported consanguinity was 16/24 (67%). The mutation detection rate in the cohort is similar to previous reports [2, 5, 4]. Using these data and dividing by the number of live births from 2007 to 2016 for England, Scotland, Wales and Northern Ireland, which was 7,892,004, the minimum annual incidence of CHI in the UK was calculated as 1 in 28,389 live births (95% CI 1:25 000, 1:32 258) (Fig 2). This decreases to 1 in 34,463 for those with CHI at 12 months of age or older. Where a disease-causing variant was identified, the incidence was 1 in 51,247 for those with CHI at 6 months of age and 1 in 54,806 for those with CHI at 12 months.

## Discussion

We have calculated an incidence of 1 case of CHI per 28,389 live births in the UK from 2007 to 2016. This estimate is similar to that previously reported for Ireland (1 in 27,000) but higher than estimates for other outbred populations (Fig 2) [6, 7, 8, 9, 5]. Although this could reflect a true increase in incidence, given the overlapping or near-overlapping 95% CI compared to estimates from other countries, the difference is more likely due to the small sample sizes in other studies (S1 Table). Regardless, this study provides an accurate estimate on the minimal incidence of CHI in the UK, which was previously unknown.

A greater incidence of CHI could be due to a number of factors. Improved awareness of CHI and a better understanding of the genetic aetiologies over the last decades is likely to have resulted in an increase in detection and reporting of CHI cases. We did not detect an overall trend of increasing incidence from 2007–2016 (S2 Table), however the increase in numbers may well have occurred prior to this time period. Differences in inclusion criteria may have also contributed with some studies using higher insulin thresholds, possibly due to differences in assay sensitivity, and this could have resulted in underdetection of CHI [9].

Furthermore, although most cases of transient CHI resolve by 6 months of age, our cohort may have included transient cases that had not yet resolved. However, given that 83% of cases continued to persist at 12 months of age (Fig 1), this is unlikely to have played a significant role. In our cohort, consanguinity was reported in 24 individuals; the pick-up rate for mutations within this group was high (16/24, 67%) in keeping with the autosomal recessive nature of the most common genetic aetiologies. This high prevalence of recessive genetic disease in CHI can explain the much higher incidence rates of CHI in consanguineous populations: 1 in 2,675 in Saudi Arabia and 1 in 19,960 in Kuwait [12, 13].

The major limitation of this study is that it is derived from a single centre to which referrals for genetic testing were voluntary. However, as CHI in the UK is managed by a highly-specialised service funded on a national level, most cases of CHI in which genetic testing would have been clinically indicated (i.e. severity, persistence of disease or features suggesting a genetic aetiology) would have been referred to the Exeter laboratory for testing. Moreover, as genetic testing is government-funded it is unlikely that patients would have been tested outside of the UK.

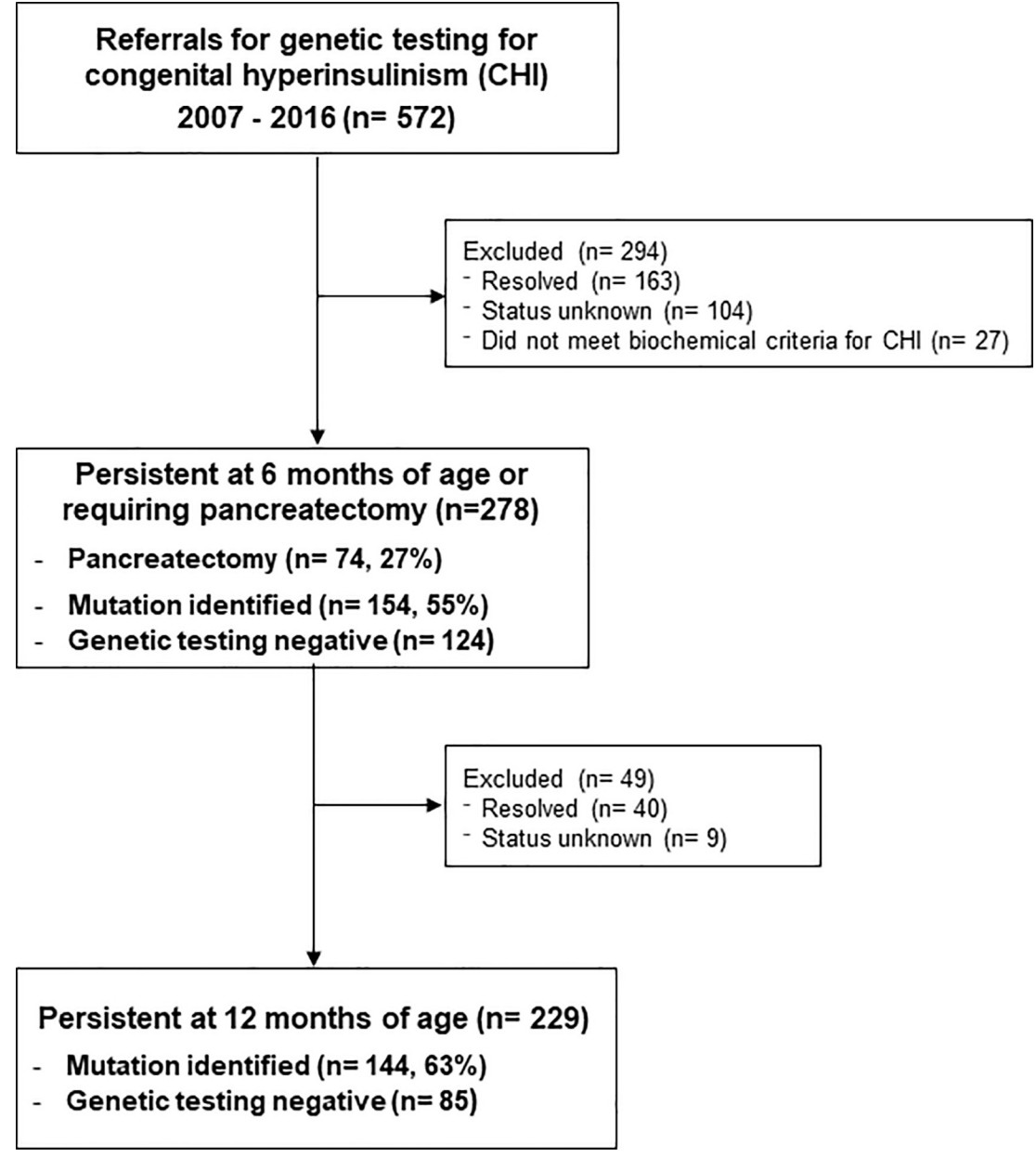

**Fig 1. Flow diagram showing the outcomes of referrals for genetic testing for congenital hyperinsulinism.**

The criteria for CHI diagnosis was a non-suppressed insulin and/or C-peptide level at the time of hypoglycaemia. While the measurement of C-peptide may not be specific to the diagnosis of CHI, its use as an adjunct with serum insulin in the context of robust supportive clinical criteria is reasonably justified and is unlikely to over-represent the reported incidence of CHI. A further limitation of our study is that we were unable to determine persistence of CHI in 104 patients. Given that these patients were lost to follow-up, it seems most likely that the CHI was transient and would therefore be unlikely to affect the calculated incidence of persistent CHI. Our calculation thus reflects the minimal incidence of CHI in the UK. It is important to note that transient cases can be severe despite the self-resolving nature and have been reported to be more common than persistent forms, consistent with the association with

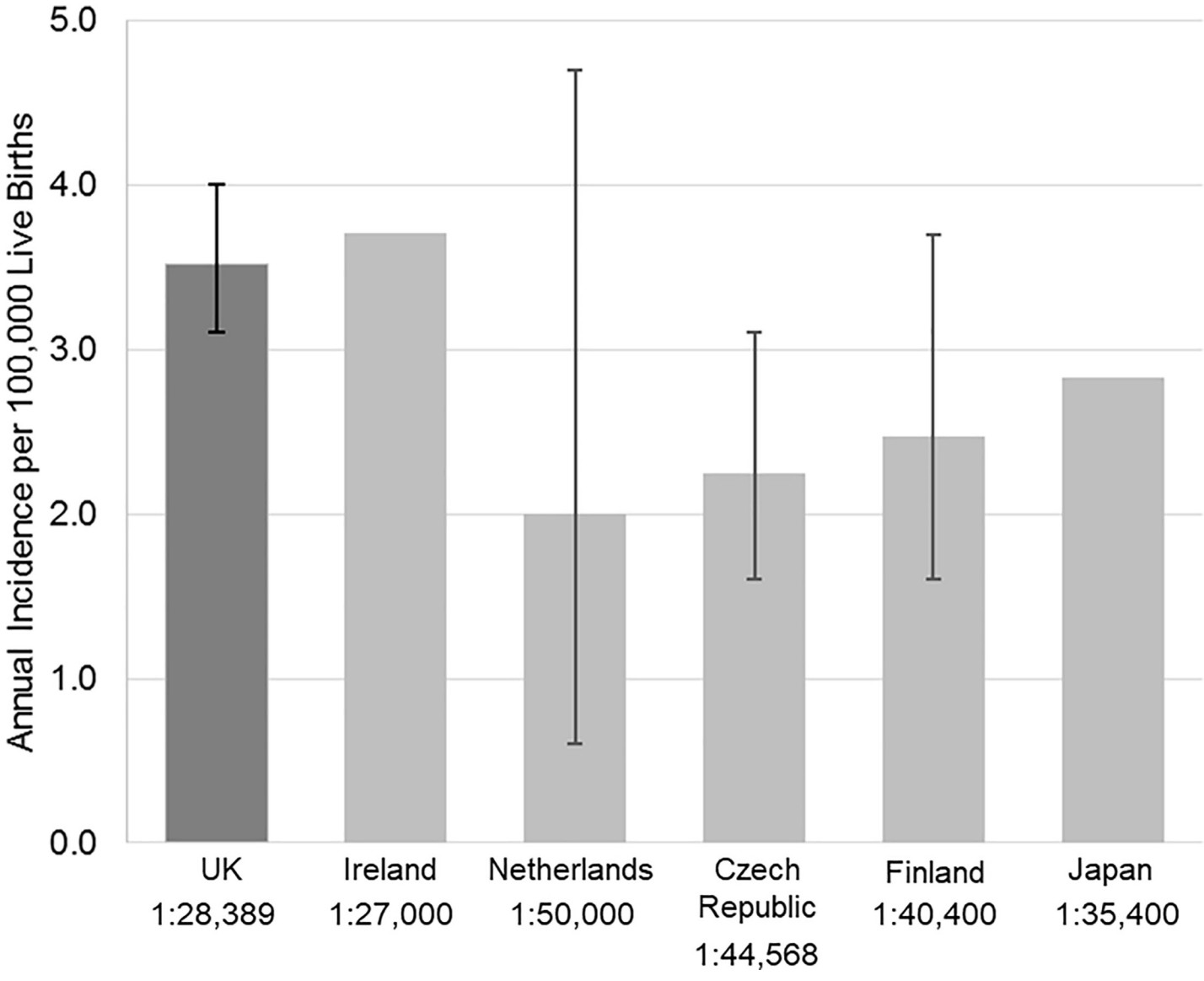

**Fig 2. The incidence of congenital hyperinsulinism in the UK compared to other outbred populations.** In the current study, CHI was diagnosed based on a non-suppressed insulin and C-peptide level at the time of hypoglycaemia with persistence beyond 6 months of age or need for pancreatectomy. Incidence is expressed as the number per 100,000 live births with the 95% confidence intervals (CI) provided with each estimate where calculated. The 95% CI could not be calculated for Ireland or Japan as the estimates were based on unpublished or unavailable data, respectively.

perinatal stress [9]. Transient CHI therefore deserves further study to establish its incidence and natural history.

Ensuring accurate calculations of the incidence of disease has important implications for both health care and medical research. The clinical management of CHI, particularly persistent forms, can be complex, requiring input from endocrinology, specialised nuclear medicine imaging, surgical expertise, allied health professionals and clinical laboratories [11]. Accurate incidence data are therefore important for health services planning. As well, hypoglycaemia can be due to causes other than CHI, and it is important to understand the incidence of CHI relative to other aetiologies. Furthermore, as genetic sequencing technologies have become more widely used both as clinical and research tools, rare genetic variants are increasingly

identified which are often difficult to classify. One widely used approach to help determine likely pathogenicity is to assess how frequently the variant appears in large population databases such as the Genome Aggregation Database (gnomAD) [14, 15]. The success of this approach however relies on accurate calculations of the incidence disease within the population.

In conclusion we have shown that the incidence of CHI in the UK is at least 1 in 28,389 live births. Current incidence estimates are imperative for both health care planning and accurate interpretation of the results of genetic testing, which will help advance our understanding of CHI.

## Supporting information

**S1 Table. Data used for CHI incidence and 95% CI calculation in other outbred study populations.**
(DOCX)

**S2 Table. Incidence of CHI in the UK by year from 2007 to 2016.** The cases of CHI had either persisted beyond 6 months of age or required pancreatectomy for hypoglycaemia.
(DOCX)

## Author Contributions

**Conceptualization:** Daphne Yau, Thomas W. Laver, Sarah E. Flanagan.

**Data curation:** Daphne Yau, Jayne A. L. Houghton, Sarah E. Flanagan.

**Formal analysis:** Daphne Yau, Thomas W. Laver.

**Funding acquisition:** Sian Ellard, Sarah E. Flanagan.

**Investigation:** Daphne Yau, Antonia Dastamani, Senthil Senniappan, Guftar Shaikh, Tim Cheetham, Talat Mushtaq, Ritika R. Kapoor, Tabitha Randell.

**Methodology:** Daphne Yau, Sarah E. Flanagan.

**Project administration:** Daphne Yau, Sarah E. Flanagan.

**Resources:** Antonia Dastamani, Senthil Senniappan, Guftar Shaikh, Tim Cheetham, Talat Mushtaq, Ritika R. Kapoor, Tabitha Randell, Pratik Shah, Indraneel Banerjee.

**Supervision:** Sian Ellard, Pratik Shah, Indraneel Banerjee, Sarah E. Flanagan.

**Validation:** Daphne Yau, Sarah E. Flanagan.

**Writing – original draft:** Daphne Yau, Sarah E. Flanagan.

**Writing – review & editing:** Daphne Yau, Thomas W. Laver, Antonia Dastamani, Jayne A. L. Houghton, Guftar Shaikh, Tim Cheetham, Talat Mushtaq, Ritika R. Kapoor, Tabitha Randell, Indraneel Banerjee, Sarah E. Flanagan.

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
