## [Decision Letter · Decision Letter 0]

18 Nov 2019

PONE-D-19-26690

Using referral rates for genetic testing to determine the incidence of a rare disease: the minimal incidence of congenital hyperinsulinism in the UK is 1 in 28,389

PLOS ONE

Dear Dr. Daphne Yau,

Thank you for submitting your manuscript to PLOS ONE. After careful consideration, we feel that it has merit but does not fully meet PLOS ONE’s publication criteria as it currently stands. Therefore, we invite you to submit a revised version of the manuscript that addresses the points raised during the review process.

The authors present a interesting study but some areas needs attention. Please, respond to th questions raised by reviewer 1. Indeed, the questions around the use of C-peptide as a diagnostic criteria is relevant and presentation of the actual measurements including p-insulin and p-glucose would be of great value. 

We would appreciate receiving your revised manuscript by Jan 02 2020 11:59PM. To enhance the reproducibility of your results, we recommend that if applicable you deposit your laboratory protocols in protocols.io, where a protocol can be assigned its own identifier (DOI) such that it can be cited independently in the future. For instructions see: http://journals.plos.org/plosone/s/submission-guidelines#loc-laboratory-protocols

We look forward to receiving your revised manuscript.

Kind regards,

Klaus Brusgaard

Academic Editor

PLOS ONE

Journal Requirements:

2. In ethics statement in the manuscript and in the online submission form, please provide additional information about the database used in your retrospective study. Specifically, please ensure that you have discussed whether all data were fully anonymized before you accessed them and/or whether the IRB or ethics committee waived the requirement for informed consent. If patients provided informed written consent to have their data used in research, please include this information.

Reviewers' comments:

Reviewer's Responses to Questions

**Comments to the Author**

1. Is the manuscript technically sound, and do the data support the conclusions?

Reviewer #1: Partly

Reviewer #2: Yes

2. Has the statistical analysis been performed appropriately and rigorously? 

Reviewer #1: No

Reviewer #2: Yes

3. Have the authors made all data underlying the findings in their manuscript fully available?

Reviewer #1: No

Reviewer #2: Yes

4. Is the manuscript presented in an intelligible fashion and written in standard English?

Reviewer #1: Yes

Reviewer #2: Yes

5. Review Comments to the Author

Reviewer #1: This is an interesting study of clinical importance, however with some epidemiological flaws.

1. In the methods, it is not clear how the clinical data were obtained. Did you have access to hospital file data or were the data based on questionnaires in the genetic testing request form? Genetic testing usually takes place prior to surgery. How did you obtain the surgery data? How did you know what age the patients had when determining persistence > 6 or 12 months?

2. Methods: An “or” diagnostic criteria of elevated C-peptide is doubtful, as the prolonged half-life of C-peptide compared to insulin implies that p-C-peptide does not reflect rapid changes in p-glucose. Indeed, the two references mentioned (Stanley, Banerjee) did not use elevated C-peptide as a diagnostic criteria. However, the inclusion criteria were sound (>6 mo CHI or surgery). However, it is very unlikely that an infant with an erroneous diagnosis of CHI based on C-peptide would be subjected to pancreatectomy or would still require therapy after 6 mo. The diagnostic problem on C-peptide should be mentioned in Limitations.

3. In Results, the only included patients had CHI persisting > 6 months’ age. Did no patient undergo pancreatic surgery <6 months’ age? And if they did, did you miss to detect them?

4. Why are the p-glucose, p-insulin and C-peptide data not presented? Is the study based on genetic testing requests only without actual data on the diagnosis? If the diagnoses were not actually validated it would be fair to state.

5. A patient inclusion flow chart is missing. The 104 patients lost to follow-up should be presented in a Results flow chart Figure 1. Why was it not possible in a country like the UK to perform a follow-up on these children?

6. An incidence calculation is always an estimate of the true, unknown incidence. Although many publications on incidences do not deal with details, a report on incidence only definitely should. For epidemiologists, a 95% confidence interval would be a demand. Your sampling is large enough to give a narrower C.I. than most others in the field. If you compute 95% C.I.s on the other incidences reported from other countries, one would better understand whether differences are significant or due to small samples.

7. Moreover, your data set is large enough to dive a little into time trends and geographical distribution. Did the incidence increase with time? This would suggest more referrals rather than a true increase. Were some geographic areas underrepresented? If yes, why?

8. In Discussion, the calculation is presented as a minimal incidence, as probably, or almost, all UK patients had genetic testing done in Exeter. Is it possible that some persistent CHI patients did not undergo genetic testing at all, perhaps especially in the beginning of the period studied?

Reviewer #2: This is a well designed and conducted epidemiological study that is presented in a clearly and concisely. The methods and interpretation of data are appropriate. The limitations noted are comprehensive and reasonable. While this is not a landmark study, it does add valuable incidence information that will be useful for planning and operationalizing treatment of a condition that affects a very small population.

6. PLOS authors have the option to publish the peer review history of their article (what does this mean?). If published, this will include your full peer review and any attached files.

Reviewer #1: No

Reviewer #2: No

---

## [Author Response · Author response to Decision Letter 0]

26 Dec 2019

The authors present a interesting study but some areas needs attention. Please, respond to th questions raised by reviewer 1. Indeed, the questions around the use of C-peptide as a diagnostic criteria is relevant and presentation of the actual measurements including p-insulin and p-glucose would be of great value. 

Thank you for your reply regarding our manuscript, “Using referral rates for genetic testing to determine the incidence of a rare disease: the minimal incidence of congenital hyperinsulinism in the UK is 1 in 28,389”. We have addressed the points raised during the review process and enclose a point-by-point response as well as revised manuscript. We hope these changes are found to be satisfactory.

Reviewer #1: This is an interesting study of clinical importance, however with some epidemiological flaws.

1. In the methods, it is not clear how the clinical data were obtained. Did you have access to hospital file data or were the data based on questionnaires in the genetic testing request form? Genetic testing usually takes place prior to surgery. How did you obtain the surgery data? How did you know what age the patients had when determining persistence > 6 or 12 months?

We thank the Reviewer for their positive and constructive comments. The clinical data were obtained from the genetic testing request form, which encompassed details regarding hyperinsulinism treatment including surgery. If the outcome was unknown, the requesting clinician was contacted to provide data regarding persistence at 6 and 12 months of age, and treatment including surgery. We have revised the manuscript to reflect this.

2. Methods: An “or” diagnostic criteria of elevated C-peptide is doubtful, as the prolonged half-life of C-peptide compared to insulin implies that p-C-peptide does not reflect rapid changes in p-glucose. Indeed, the two references mentioned (Stanley, Banerjee) did not use elevated C-peptide as a diagnostic criteria. However, the inclusion criteria were sound (>6 mo CHI or surgery). However, it is very unlikely that an infant with an erroneous diagnosis of CHI based on C-peptide would be subjected to pancreatectomy or would still require therapy after 6 mo. The diagnostic problem on C-peptide should be mentioned in Limitations.

We acknowledge the Reviewer’s point regarding the use of C-peptide in diagnosing hyperinsulinism. We used C-peptide as an additional biomarker in the context of recurrent, severe hypoglycaemia and high glucose infusion rates consistent with hyperinsulinism causing hypoglycaemia while recognizing the importance of insulin as a prime diagnostic marker. We have revised the manuscript to reflect the Reviewer’s comments.

3. In Results, the only included patients had CHI persisting > 6 months’ age. Did no patient undergo pancreatic surgery <6 months’ age? And if they did, did you miss to detect them?

We thank the Reviewer for clarifying this. The patients undergoing surgery were included in N=278 patients. We apologise for not having made this clearer and have amended the manuscript to indicate the number of patients undergoing pancreatic surgery. 

4. Why are the p-glucose, p-insulin and C-peptide data not presented? Is the study based on genetic testing requests only without actual data on the diagnosis? If the diagnoses were not actually validated it would be fair to state.

The genetic testing request form included data on glucose, insulin and c-peptide to confirm evidence of CHI prior to genetic testing. We have now included these values in the results of the manuscript.

5. A patient inclusion flow chart is missing. The 104 patients lost to follow-up should be presented in a Results flow chart Figure 1. Why was it not possible in a country like the UK to perform a follow-up on these children?

A patient inclusion flow chart was provided as a supplemental figure. We have now provided this as figure 1 and the incidence data as figure 2. For the 104 patients lost to follow up, we had contacted the referring physicians to obtain outcome data, but either the patient was no longer in the physician’s care or were lost to their follow up, or we did not receive a reply despite several attempts. 

6. An incidence calculation is always an estimate of the true, unknown incidence. Although many publications on incidences do not deal with details, a report on incidence only definitely should. For epidemiologists, a 95% confidence interval would be a demand. Your sampling is large enough to give a narrower C.I. than most others in the field. If you compute 95% C.I.s on the other incidences reported from other countries, one would better understand whether differences are significant or due to small samples.

We have now calculated 95% confidence intervals for our data set as well as for the incidences reported from other countries. These are included these in figure 2 and the data used for calculation in supporting table S1. However, we were unable to calculate 95% CI for the incidences for Ireland and Japan. The former is based on unpublished data and the latter is published in Japanese. Overall, the estimates from the outbred populations are overlapping or nearly overlapping, in the case of the Czech Republic estimate. It therefore appears the differences are due to small sample sizes in the other populations versus a significant differences between the populations. We have revised our discussion to reflect this analysis.

7. Moreover, your data set is large enough to dive a little into time trends and geographical distribution. Did the incidence increase with time? This would suggest more referrals rather than a true increase. Were some geographic areas underrepresented? If yes, why?

As the reviewer has suggested, we have examined the incidence over time (S2 table). Although the annual incidence fluctuated year by year, a sustained increase was not observed over the time period studied. We have not examined the incidence by geography as we do not have data on the location of birth, only location of treatment. It would not be accurate to include data on treatment location given that there are two tertiary referral centres to which patients are referred from all over the UK. 

8. In Discussion, the calculation is presented as a minimal incidence, as probably, or almost, all UK patients had genetic testing done in Exeter. Is it possible that some persistent CHI patients did not undergo genetic testing at all, perhaps especially in the beginning of the period studied?

We anticipate most if not all patients undergoing genetic testing for CHI in the UK would have been referred to Exeter. However, it is possible, as the reviewer states, that some patients with CHI may not have undergone testing at all, particularly if the hypoglycaemia could be managed with diazoxide. 

Reviewer #2: This is a well designed and conducted epidemiological study that is presented in a clearly and concisely. The methods and interpretation of data are appropriate. The limitations noted are comprehensive and reasonable. While this is not a landmark study, it does add valuable incidence information that will be useful for planning and operationalizing treatment of a condition that affects a very small population.

We kindly thank the Reviewer for their positive comments regarding our study.

---

## [Decision Letter · Decision Letter 1]

15 Jan 2020

Using referral rates for genetic testing to determine the incidence of a rare disease: the minimal incidence of congenital hyperinsulinism in the UK is 1 in 28,389

PONE-D-19-26690R1

Dear Dr. Daphne Yau,

We are pleased to inform you that your manuscript has been judged scientifically suitable for publication and will be formally accepted for publication once it complies with all outstanding technical requirements.

With kind regards,

Klaus Brusgaard

Academic Editor

PLOS ONE

Additional Editor Comments (optional):

Reviewers' comments:

Reviewer's Responses to Questions

**Comments to the Author**

1. If the authors have adequately addressed your comments raised in a previous round of review and you feel that this manuscript is now acceptable for publication, you may indicate that here to bypass the “Comments to the Author” section, enter your conflict of interest statement in the “Confidential to Editor” section, and submit your "Accept" recommendation.

Reviewer #1: All comments have been addressed

Reviewer #2: All comments have been addressed

2. Is the manuscript technically sound, and do the data support the conclusions?

Reviewer #1: Yes

Reviewer #2: (No Response)

3. Has the statistical analysis been performed appropriately and rigorously? 

Reviewer #1: Yes

Reviewer #2: (No Response)

4. Have the authors made all data underlying the findings in their manuscript fully available?

Reviewer #1: Yes

Reviewer #2: (No Response)

5. Is the manuscript presented in an intelligible fashion and written in standard English?

Reviewer #1: Yes

Reviewer #2: (No Response)

6. Review Comments to the Author

Reviewer #1: The authors have responded well to the questions raised. A few typos remains:

Line 109: "This data" should be "These data"

Line 114: "incidence 1 in 51,247" should be "incidence was 1 in 51,247"

Line 144: "under detection" should be "underdetection"

Line 180: "incidence data is" should be "incidence data are"

Reviewer #2: (No Response)

7. PLOS authors have the option to publish the peer review history of their article (what does this mean?). If published, this will include your full peer review and any attached files.

Reviewer #1: Yes: Henrik Thybo Christesen

Reviewer #2: Yes: Trevor Richter

---

## [Editor Report · Acceptance letter]

21 Jan 2020

PONE-D-19-26690R1 

Using referral rates for genetic testing to determine the incidence of a rare disease: the minimal incidence of congenital hyperinsulinism in the UK is 1 in 28,389 

Dear Dr. Yau:

I am pleased to inform you that your manuscript has been deemed suitable for publication in PLOS ONE. Congratulations! Your manuscript is now with our production department. 

With kind regards,

on behalf of

Dr. Klaus Brusgaard 

Academic Editor

PLOS ONE